# Engaging Leadership Reduces Quiet Quitting and Improves Work Engagement: Evidence from Nurses in Greece

**DOI:** 10.3390/nursrep15070247

**Published:** 2025-07-04

**Authors:** Ioannis Moisoglou, Aglaia Katsiroumpa, Ioanna V. Papathanasiou, Olympia Konstantakopoulou, Maria Katharaki, Maria Malliarou, Konstantinos Tsaras, Ioanna Prasini, Maria Rekleiti, Petros Galanis

**Affiliations:** 1Department of Nursing, University of Thessaly, 41500 Larissa, Greece; iopapathanasiou@uth.gr (I.V.P.); malliarou@uth.gr (M.M.); ktsa@uth.gr (K.T.); 2Clinical Epidemiology Laboratory, Faculty of Nursing, National and Kapodistrian University of Athens, 11527 Athens, Greece; aglaiakat@nurs.uoa.gr (A.K.); pegalan@nurs.uoa.gr (P.G.); 3Center for Health Services Management and Evaluation, Faculty of Nursing, National and Kapodistrian University of Athens, 11527 Athens, Greece; olykonstant@nurs.uoa.gr (O.K.); mkatharaki@nurs.uoa.gr (M.K.); 4Palliative Care Galilee, 19004 Spata, Greece; iprasini@galilee.gr; 5Andreas Syggros Hospital of Cutaneous and Venereal Diseases, 16121 Athens, Greece; proistep@syggros.gr

**Keywords:** engaging leadership, leadership, nurses, quiet quitting, work, work engagement

## Abstract

**Background:** The leadership style employed by those in positions with authority could significantly impact employees’ work behavior, either positively or negatively. **Objectives**: The aim of the study was to examine the impact of engaging leadership on quiet quitting and work engagement among nurses. **Methods:** A cross-sectional study was conducted in Greece with a convenience sample. Data collection occurred throughout October 2024 via an online survey. We utilized Google forms to create an online version of the study questionnaire, which was subsequently shared in Facebook groups for nurses. We used validated tools to measure our study variables, namely, the “Engaging Leadership Scale-12” for engaging leadership, the “Quiet Quitting Scale” for quiet quitting, and the “Utrecht Work Engagement Scale-3” for work engagement. We applied univariate and multivariable linear regression analyses to identify the association between engaging leadership, quiet quitting, and work engagement. *p*-values less than 0.05 were considered statistically significant. **Results:** Our multivariable models showed a negative association between engaging leadership and quiet quitting. Moreover, we found a positive association between engaging leadership and work engagement after the elimination of confounders. **Conclusions**: Our findings suggest that higher levels of engaging leadership reduce quiet quitting and improve work engagement. Nurses’ managers should adopt engaging leadership to improve nurses’ motivation and, thus, clinical outcomes.

## 1. Introduction

The nursing profession entails a particularly demanding working environment, as nurses must address substantial workloads, cater to the escalating demands of patients with chronic illnesses, and fulfill the imperatives of enhanced efficiency alongside the safety and quality of healthcare services [1]. Nurses frequently lack the necessary resources to effectively meet these demands, as their working environment is characterized by understaffing, an insufficient provision of essential resources, and a lack of organizational support, making their work emotionally and physically demanding [2,3]. This disparity between job demands and resources may result in nurses’ disengagement and quiet quitting [4,5]. Supervisors can improve nurses’ work engagement by implementing an appropriate leadership style, which will inspire, empower, and motivate them.

According to Schaufeli et al. [6] engagement is defined as “*a positive, fulfilling, work-related state of mind that is characterized by vigor, dedication, and absorption*”. Engaged employees exhibit elevated energy levels, a propensity for increased effort, and, when confronted with challenges, the mental resilience to persevere. They are fervent about their job, perceive it as significant, take pride in it, and draw inspiration from it. Their dedication to their task is so strong that time passes quickly, making it increasingly difficult to detach [7]. Operating in a supportive environment and having the necessary resources to carry out their duties boosts nurses’ job engagement [8]. Specifically, fostering collaboration and teamwork among health professionals, facilitating social support from colleagues and supervisors, enhancing structural empowerment and autonomy, offering opportunities for growth and development, ensuring high motivation and the availability of incentives along with recognizing their contributions, are elements that positively impact nurses’ work engagement [9,10,11]. The leadership style adopted by supervisors is equally crucial in fostering nurses’ work engagement. By embracing the transformational leadership style, supervisors articulate a future vision, devise implementation strategies, and empower their subordinates by their conduct. Simultaneously, they foster creativity and innovation among their followers, address developmental needs, and provide support and coaching for their growth [12]. In response to this leadership behavior, followers strive to align themselves with the leader, thereby enhancing their work engagement [8,13]. The advantages of nurses’ work engagement are numerous and impact nursing personnel, the organization, and patients. Nurses who are engaged perform better at work, have higher job satisfaction, and exhibit a reduced likelihood of job turnover, while patients hospitalized in wards with engaged nurses receive high-quality and safe healthcare services [9,14,15]. Work engagement plays an important role in both positive and negative work experiences for nurses. When nurses do experience resilience, through work engagement they achieve a greater level of performance, and similarly, when they are experiencing psychological distress, work engagement contributes to increased job satisfaction [16,17].

The term “quiet quitting” gained prominence in 2022 via a brief video on TikTok, in which the creator elucidates the phenomenon of quiet quitting within the context of contemporary work culture [18]. The term “quiet quitting” emerged as a persistent work behavior rather than a fleeting social media trend, having been embraced by several employees across various sectors. A Gallup study revealed that 50% of employees in the business sector in United States opt for quiet quitting [19]. According to the findings of the same study, employees who choose quiet quitting provide the minimum services, barely avoid being fired, and are not going above and beyond at work; they are just meeting their job description and are psychologically detached from their job [19]. Hungerford and her colleagues characterize quiet quitting as a manifestation of disengagement, wherein the quiet quitters remain passive, refraining from exceeding their contractual obligations or exerting additional effort, even when presented with financial incentives [20]. Quiet quitting may be perceived as a self-protective stance adopted by employees. Due to escalating job expectations in a competitive work environment and a lack of personal and growth opportunities, many prefer to sustain their employment by engaging in quiet quitting to preserve their work–life balance [18].

Gallup recognizes that management plays a crucial role in the phenomenon of quiet quitting, asserting that it is indicative of poor management practices [19]. In the healthcare sector, nurses commit quiet quitting at a greater rate than other health professionals [21]. Organizational support for nurses may adversely impact quiet quitting, whereas burnout and dissatisfaction exacerbate this behavior [22,23]. Additional risk factors for quiet quitting include workplace bullying, excessive workload, and aspects of the work environment such as diminished nurse involvement in hospital matters, weaker nurse–physician relationships, inadequate nursing foundations for quality care, and reduced competencies in nurse management, leadership, and support [4,24,25].

Changes in the working conditions of nurses, who are frontline healthcare professionals, have put the healthcare industry at a crucial crossroads. On the one hand, a lot of nurses are leaving their professions, and on the other hand, a sizable portion of nurses who stay within the profession either opt to quiet quitting or exhibit extremely low levels of engagement. To help administrators of health organizations and those involved in the planning of health policies create a healthy work environment that will attract and motivate nurses, it is imperative that all work-related factors that contribute to the above behaviors among nurses be examined and highlighted.

Engaging leadership is a contemporary concept established to examine how leaders fulfill the basic needs of their followers [26]. Engaging leadership is grounded in Self-Determination Theory, which posits that individuals are significantly motivated towards personal development and psychological well-being when three fundamental needs are satisfied: autonomy, competence, and relatedness. The need for autonomy is fulfilled when individuals are afforded the ability to make their own decisions. The need for competence is fulfilled when individuals see their ability to navigate the environment, whereas the demand for relatedness arises from the aspiration to cultivate affirmative connections with others [27]. Research, beyond the healthcare delivery sector, has emphasized the direct positive influence of engaging leadership on work engagement, as well as the indirect influence through the fulfillment of fundamental psychological needs, the availability of job resources, and the effect on perceived intrinsic organizational values [28,29].

The healthcare sector is at a critical crossroads due to changes in nurses’ work behavior, who serve as frontline healthcare professionals. On the one hand, we have a large number of nurses leaving the profession [30], and on the other hand, a significant number of nurses who remain in the profession and show very low levels of engagement or choose quiet quitting. Such work behaviors may compromise the quality of care provided [31]. Leadership style can positively or negatively influence nurses’ work behavior. The relationship between leadership and work engagement among nurses is well established [32]. Although the literature on quiet quitting has identified factors that lead nurses to choose this work behavior [33], there is a gap in the literature regarding the effect of leadership, and in particular engaging leadership, on the quiet quitting of nurses. Identifying all organizational factors that may influence work engagement and quiet quitting will provide valuable information to both health organization managers and those who design human resource policies to create a healthy, supportive work environment that motivates nursing staff.

As there is a gap in the literature regarding engaging leadership and nursing staff, we performed a study to explore for the first time the impact of engaging leadership on quiet quitting among nurses. Additionally, we explored the impact of engaging leadership on nurses’ work engagement, and to the best of our knowledge, our study is the second to examine the effect of engaging leadership on nurses work engagement [34].

## 2. Materials and Methods

### 2.1. Design and Sampling

A cross-sectional study was undertaken using a sample of Greek nurses. Data collection occurred throughout October 2024 via an online survey. Specifically, we utilized Google forms to create an online version of the study questionnaire, which was subsequently shared in Facebook groups for nurses. These groups include approximately 2000 nurses aged 22 to 70 years old that have been working in clinical settings. The purpose of these groups is to provide information on work-related issues regarding nursing, such as scientific articles, laws in healthcare services and nursing, and employee issues. Nurses from anywhere in Greece may apply to be a member of these groups. Administrators of the groups examine the application and control membership. This method yielded a convenience sample of nurses. The inclusion criteria were as follows: (a) nurses employed in clinical environments, (b) nurses with a minimum of one year’s work experience, and (c) nurses proficient in Greek; (d) participation was voluntary. Leader nurses such as head nurses and department head nurses could not participate in our study since their work status differed from nurse employees. Also, nurses that did not work in clinical settings (e.g., school nurses) were excluded from our study.

We employed G*Power version 3.1.9.2. for sample size calculation. In particular, we considered (a) a low effect size (f^2^ = 0.05) between engaging leadership, quiet quitting and work engagement, (b) the number of independent variables (four predictors and five confounders), (c) a confidence level of 95%, and (d) a margin of error of 1%. Overall, our sample size was estimated at 370 nurses.

### 2.2. Instruments

We measured demographic data of nurses by asking them several questions before they answered our study scales. Regarding demographic variables we measured sex (females or males), age (continuous variable), work in an understaffed ward (no or yes), shift work (no or yes), and work experience (continuous variable). Also, we asked participants if they work as head nurses (no or yes), and if they work in clinical settings (no or yes). Thus, we included seven items for demographic variables. Applying our inclusion and exclusion criteria, head nurses and those who do not work in clinical settings were removed from further statistical analysis.

We used the “Engaging Leadership Scale-12” (ELS-12) to measure levels of engaging leadership among our nurses. The initial version of the questionnaire contained nine items divided into three factors (inspiring, strengthening, and connecting) [26] and in the final version the Empowering factor was added with three items as well [35]. Schaufeli developed the ELS-12 in 2021. The ELS-12 comprises 12 items that represent four factors of engaging leadership, namely, strengthening, connecting, empowering, and inspiring. Nurses answer the ELS-12 by thinking of their own department’s supervisor. Example items on the ELS-12 include the following: “My supervisor encourages team members to develop their talents as much as possible”, “My supervisor encourages collaboration among team members”, and “My supervisor gives team members enough freedom and responsibility to complete their tasks”. Answers are recorded on a five-point Likert scale from completely disagree (1) to completely agree (5). Scores on each factor range from 1 to 5. Higher values indicate higher levels of engaging leadership. We used the validated Greek version of the ELS-12. Katsiroumpa et al. validated the ELS-12 in Greek and tested its reliability in 2024 [36]. In our study, Cronbach’s alpha for the factors “strengthening”, “connecting”, “empowering”, and “inspiring” was 0.828, 0.920, 0.867, and 0.936, respectively.

To assess quiet quitting levels among the nurses, we employed the “Quiet Quitting Scale” (QQS) [37]. Galanis et al. developed the QQS in 2023. The instrument comprises nine items, with responses recorded on a five-point Likert scale ranging from strongly disagree/never (1) to strongly agree/always (5). Sample items include “I take as many breaks as I can”, “I don’t express opinions and ideas about my work because I think that work conditions are not going to change”, and “If a colleague can do some of my work, then I let him/her do it”. The QQS includes three factors, i.e., “detachment” (four items), “lack of initiative” (three items), and “lack of motivation” (two items). Scores on each factor are calculated as the mean of responses to the factor items, resulting in a range from 1 to 5. Higher scores indicate greater levels of quiet quitting. We applied the recommended cut-off point of 2.06 to categorize nurses as quiet quitters or non-quiet quitters [38]. The validated Greek version of the QQS was used. Galanis et al. validated the QQS in Greek and tested its reliability in 2024 [21]. In our study, Cronbach’s alpha for the factors “detachment”, “lack of initiative”, and “lack of motivation” were 0.771, 0.729, and 0.786, respectively.

Work engagement in our sample was measured using the “Utrecht Work Engagement Scale-3” (UWES-3) [39]. The scale consists of three items (e.g., “At my work, I feel bursting with energy”), with responses recorded on a seven-point Likert scale from never (0) to every day (6). The mean UWES-3 score ranges from 0 to 6, with higher values indicating increased work engagement. We utilized the validated Greek version of the UWES-3 [40]. In our study, the Cronbach’s alpha for the UWES-3 was 0.812.

### 2.3. Ethical Considerations

Our study protocol received approval from the Ethics Committee of the Faculty of Nursing at the National and Kapodistrian University of Athens (approval number; 01, 26 September 2024). We also conducted our research in accordance with the Declaration of Helsinki [41]. Participants were informed about the study’s aim and design through an information sheet on the online questionnaire. Consent was obtained before participation, and only nurses who agreed were allowed to proceed with answering the questionnaire.

### 2.4. Data Analysis

For data presentation, we use numbers and percentages for categorical variables, while continuous variables are described using mean, standard deviation (SD), median, minimum value, and maximum value. The Kolmogorov–Smirnov test and Q-Q plots were employed to assess the distribution of continuous variables. Our independent variables included the four factors of the ELS-12, namely, strengthening, connecting, empowering, and inspiring. Also, scores on the three factors of the QQS (i.e., “detachment”, “lack of initiative”, and “lack of motivation”), and the UWES-3 were the dependent variables. Our dependent variables followed a normal distribution, and thus we performed a linear regression analysis. We began with a univariate linear regression analysis, followed by the creation of a final multivariable linear regression model incorporating all independent variables. The multivariable models were adjusted for sex, age, understaffed ward, shift work, and work experience. We report unadjusted and adjusted beta coefficients, 95% confidence intervals (CI), and *p*-values, with statistical significance set at *p* < 0.05. A statistical analysis was performed using IBM SPSS 21.0 (International Business Machines Corporation Statistical Package for the Social Sciences).

## 3. Results

### 3.1. Demographics

Our sample included 404 nurses. The majority of nurses were females (88.9%, n = 359), while 11.1% (n = 45) were males. In our sample, the mean age was 41.1 ± 9.9 years (min, max: 23, 61 years). Among our nurses, 82.2% (n = 332) have been working on understaffed wards, while 71.8% (n = 290) have been working on shifts. The mean years of clinical experience was 16.7 ± 14.0 years (min, max: 0.5, 40 years).

### 3.2. Study Scales

Descriptive statistics for the study scales are shown in Table 1. Regarding engaging leadership, scores on empowering (mean; 3.04 ± 1.01, min–max; 1–5), connecting (mean; 2.96 ± 1.10, min–max; 1–5), and strengthening (mean; 2.90 ± 0.98, min–max; 1–5) were higher than inspiring (mean; 2.50 ± 1.08, min–max; 1–5).

Regarding quiet quitting, the mean score on factor “lack of motivation” (mean; 2.94 ± 1.00, min–max; 1–5) was higher than the mean score on factors “lack of initiative” (mean; 2.38 ± 0.92, min–max; 1–5), and “detachment” (mean; 2.14 ± 0.78, min–max; 1–4.75). In our sample, 66.6% (n = 269) were considered as quiet quitters, while 33.4% (n = 135) were considered as non-quiet quitters.

The mean score on UWES-3 was 3.52 ± 1.60 (min–max; 0–6), indicating a moderate level of work engagement.

### 3.3. Impact of Engaging Leadership on Quiet Quitting

A univariate linear regression analysis identified a statistically significant association between all dimensions of engaging leadership and detachment, lack of initiative, and lack of motivation. However, our multivariable models showed that connecting and inspiring had an impact on quiet quitting. In particular, we found a negative association between connecting and detachment (adjusted beta = −0.215, 95% CI = −0.365 to −0.065, *p*-value = 0.005), and lack of initiative (adjusted beta = −0.199, 95% CI = −0.364 to −0.034, *p*-value = 0.018). A similar negative association was identified between inspiring and detachment (adjusted beta = −0.146, 95% CI = −0.289 to −0.004, *p*-value = 0.043), and lack of motivation (adjusted beta = −0.233, 95% CI = −0.415 to −0.052, *p*-value = 0.012). Table 2 presents a linear regression analysis with quiet quitting as the dependent variable.

### 3.4. Impact of Engaging Leadership on Work Engagement

Our univariate linear regression models showed statistically significant associations between all dimensions of engaging leadership and work engagement. After the elimination of confounders, we found that inspiring was associated with work engagement (adjusted beta = 0.400, 95% CI = 0.114 to 0.685, *p*-value = 0.0065). Table 3 presents the results from a linear regression analysis with work engagement as the dependent variable.

## 4. Discussion

This study emphasized the elevated prevalence of quiet quitting among nurses, the medium degree of work engagement, and the influence of engaging leadership on both work engagement and quiet quitting. These findings align with previous studies indicating elevated rates of quiet quitting among nurses and their modest levels of work engagement [21,42]. Key organizational antecedents that bolster nurses’ work engagement include structural empowerment, social support, and a sense of community belonging [43]. Through structural empowerment, nursing administration systems create a work environment based on necessary information and knowledge sharing, learning and development opportunities, support, and resources. Among the positive outcomes of such a work environment is the work engagement of nurses [44]. The additional resources that an engaged leader can provide are role clarity, performance feedback, opportunities for growth and development, and value congruence. Securing these job resources positively influences work engagement, since they mediate the relationship between engaging leadership and work engagement [34]. Work engagement has a beneficial function for nurses by enhancing job satisfaction, compassion satisfaction, and overall well-being, while simultaneously decreasing burnout, compassion fatigue, and the intention to resign. Organizational advantages stemming from nurses’ work engagement encompass enhanced productivity and efficiency, as well as the assurance of patient satisfaction and the quality and safety of nursing care [11].

The COVID-19 pandemic posed significant challenges to healthcare organizations due to the mass hospital admissions of several critically ill individuals. The preexisting organizational deficiencies, such as understaffing and insufficient support, resulted in a challenging working environment for nurses. Nursing understaffing and insufficient resources availability contribute to nurse burnout, which subsequently influences the decision for quiet quitting [22,45]. The resumption of standard healthcare operations revealed that nurses remained in an unsupportive work environment, leading many to either resign or persist with diminished work engagement, opting for quiet quitting [4]. Nonetheless, many who opt for quiet quitting still harbor the intention to resign from their positions [46]. The nature of nursing necessitates vigilance, continual decision-making, and collaboration with colleagues and other healthcare professionals, which starkly contrasts with the concept of quiet quitting. Thus, quiet quitting may jeopardize nursing team cohesion and the quality of care. Support from supervisors for innovation can inspire nurses and mitigate quiet quitting by encompassing idea generation, idea exploration, idea dissemination, the initiation of implementation activities, the involvement of others, and overcoming obstacles [47]. Furthermore, nurse manager ability, leadership, and support for nurses are key variables in the work environment of nurses with dual roles, as they contribute to reducing quiet quitting and strengthening their work engagement [3]. Organizational support for employees and the sustaining of a healthy work environment, irrespective of the industry, can enhance employee well-being and hence mitigate quiet quitting. Study among academics indicated that work engagement, when coupled with psychological empowerment, equitable rewards, and intrinsic motivation, positively influences work engagement and job satisfaction, hence diminishing the intention to quiet quitting [48].

A recent report by the European Observatory on Health Systems and Policies outlines six policy action goals to enable European nurses to remain in the profession [49]. These actions encompass the establishment of appealing and supportive work environments that empower nurses, ensure effective communication, foster collaborative relationships and decision-making, and provide nurses with access to information, resources, support, and opportunities for professional development [49]. Another suggested action is enhancing nursing leadership, which will be accountable for establishing these supportive work conditions [49]. The aforementioned recommended supportive actions form the core of engaging leadership, prioritizing the deployment of engaging leadership style within the management of healthcare companies, particularly in nursing administration [26]. The management of health organizations and nursing services frequently focuses on service delivery outcomes, neglecting the fact that these outcomes are actually achieved by health professionals, specifically by front-line personnel, including nurses [50]. Administrations frequently confine themselves to executing bureaucratic protocols, neglecting the well-being of human resources [51]. The report by the European Observatory on Health Systems and Policies highlights the necessity for health systems to put the emphasis on nurses and addresses the deficiency of leadership training throughout nurses’ careers [49].

As this study is the first to investigate the relationship between engaging leadership and quiet quitting, further studies in other countries and different work environments are certainly required to confirm the results of this study. In the present study, the vast majority of participants reported working in understaffed departments and on rotating shifts. Therefore, studies in better working environments are needed to investigate the degree of influence of these confounding factors on nurses’ work engagement and quiet quitting.

### Limitations

Our study faced various limitations. Primarily, data collection occurred through an online survey, resulting in a convenience sample that cannot be considered representative of Greek nurses, thus limiting the generalizability of our results. Additionally, although we employed validated instruments to assess engaging leadership, quiet quitting, and work engagement, their self-reported nature may introduce information bias. Furthermore, while our multivariable models accounted for several confounding factors, other variables could potentially influence the relationship between engaging leadership, quiet quitting, and work engagement. Finally, the study’s cross-sectional design precluded the establishment of causal relationships between the independent and dependent variables examined.

Regarding the strengths of the study, this is the first study to explore the relationship between engaging leadership, a relatively recent leadership concept, and the phenomenon of quiet quitting, which seems to be gaining ground in the nursing work environment. It is also the second study that examined the effect of engaging leadership on nurses’ work engagement. Undoubtedly, more studies are needed to confirm the findings of this study. In this context, we covered the requirements for the sample size to reduce random errors. Moreover, we used validated instruments to measure engaging leadership (i.e., Engaging Leadership Scale-12), quiet quitting (i.e., the Quiet Quitting Scale), and work engagement (i.e., Utrecht Work Engagement Scale-3). Additionally, we performed a multivariable linear regression analysis to eliminate several confounders, and thus estimate the independent effect of engaging leadership on quiet quitting and work engagement.

## 5. Conclusions

This study emphasized the impact of engaging leadership on nurses’ work engagement and quiet quitting. Human resources, particularly nurses as frontline professionals, constitute the primary asset of healthcare organizations. The efficacy of organizations is contingent upon their contributions, whereas a deficiency in work engagement and the phenomenon of quiet quitting detracts from their efficacy. Supervisors play a crucial role by employing engaging leadership to secure necessary resources and cultivate an appropriate organizational environment that addresses nurses’ needs, thereby enhancing their motivation, increasing work engagement, and mitigating the occurrence of quiet quitting.

## Figures and Tables

**Table 1 nursrep-15-00247-t001:** Descriptive statistics for the study scales.

Scales *Factors*	Mean	Standard Deviation	Median	Minimum Value	Maximum Value
**Engaging Leadership Scale-12**					
- Strengthening	2.90	0.98	3.00	1.00	5.00
- Connecting	2.96	1.10	3.00	1.00	5.00
- Empowering	3.04	1.01	3.00	1.00	5.00
- Inspiring	2.50	1.08	2.33	1.00	5.00
**Quiet Quitting Scale**					
- Detachment	2.14	0.78	2.00	1.00	4.75
- Lack of initiative	2.38	0.92	2.33	1.00	5.00
- Lack of motivation	2.94	1.00	3.00	1.00	5.00
**Utrecht Work Engagement Scale-3**	3.52	1.60	3.67	0.00	6.00

**Table 2 nursrep-15-00247-t002:** Linear regression analysis with quiet quitting as the dependent variable.

Independent Variables	Detachment						Lack of Initiative						Lack of Motivation					
	Univariate Models			Multivariable Model ^a,b^			Univariate Models			Multivariable Model ^a,c^			Univariate Models			Multivariable Model ^a,d^		
	Unadjusted Coefficient Beta	95% CI for Beta	*p*-Value	Adjusted Coefficient Beta	95% CI for Beta	*p*-Value	Unadjusted Coefficient Beta	95% CI for Beta	*p*-Value	Adjusted Coefficient Beta	95% CI for Beta	*p*-Value	Unadjusted Coefficient Beta	95% CI for Beta	*p*-Value	Adjusted Coefficient Beta	95% CI for Beta	*p*-Value
Strengthening	−0.221	−0.296 to −0.147	<0.001	0.043	−0.105 to 0.192	0.566	−0.348	−0.433 to −0.264	<0.001	−0.060	−0.223 to 0.104	0.473	−0.340	−0.434 to −0.245	<0.001	−0.078	−0.267 to 0.111	0.418
Connecting	−0.235	−0.300 to −0.169	<0.001	−0.215	−0.365 to −0.065	0.005	−0.332	−0.407 to −0.258	<0.001	−0.199	−0.364 to −0.034	0.018	−0.303	−0.387 to −0.219	<0.001	−0.042	−0.233 to 0.149	0.665
Empowering	−0.189	−0.262 to −0.116	<0.001	0.095	−0.038 to 0.228	0.160	−0.319	−0.402 to −0.237	<0.001	−0.002	−0.148 to 0.144	0.978	−0.317	−0.409 to −0.225	<0.001	−0.019	−0.189 to 0.150	0.825
Inspiring	−0.229	−0.297 to −0.162	<0.001	−0.146	−0.289 to −0.004	0.043	−0.340	−0.416 to −0.264	<0.001	−0.106	−0.263 to 0.050	0.183	−0.347	−0.431 to −0.262	<0.001	−0.233	−0.415 to −0.052	0.012

^a^ Multivariable models are adjusted for sex, age, understaffed ward, shift work, and work experience. ^b^ R^2^ for the multivariable model = 13.7%, *p*-value for ANOVA < 0.001. ^c^ R^2^ for the multivariable model = 23.4%, *p*-value for ANOVA < 0.001. ^d^ R^2^ for the multivariable model = 14.3%, *p*-value for ANOVA < 0.001. CI: confidence interval.

**Table 3 nursrep-15-00247-t003:** Linear regression analysis with work engagement as the dependent variable.

Independent Variables	Univariate Model			Multivariable Model ^a,b^		
	Unadjusted Coefficient Beta	95% CI for Beta	*p*-Value	Adjusted Coefficient Beta	95% CI for Beta	*p*-Value
Strengthening	0.565	0.415 to 0.714	<0.001	0.076	−0.221 to 0.374	0.615
Connecting	0.515	0.383 to 0.648	<0.001	0.076	−0.225 to 0.377	0.619
Empowering	0.554	0.409 to 0.698	<0.001	0.075	−0.191 to 0.342	0.580
Inspiring	0.594	0.461 to 0.727	<0.001	0.400	0.114 to 0.685	0.006

CI: confidence interval. ^a^ Multivariable linear regression model is adjusted for sex, age, understaffed ward, shift work, and work experience. ^b^ R^2^ for the multivariable model = 16.4%, *p*-value for ANOV.

## Data Availability

The data presented in this study are available at: https://doi.org/10.6084/m9.figshare.28344134.v1 (accessed on 4 February 2025).

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
