# Peer review of "Engaging Leadership Reduces Quiet Quitting and Improves Work Engagement: Evidence from Nurses in Greece"

_nursrep, 2025, doi:10.3390/nursrep15070247_

Round 1

Reviewer 1 Report

Comments and Suggestions for Authors

The manuscript addresses a topic of interest to the work environment of nurses. It concludes that engaging leadership reduces quiet quitting and improves work engagement among nurses, which is a significant finding.

While not ideal for generalizability, the approach is according to exploratory or early-stage studies and is appropriate given the study’s novelty. 

Perhaps table formatting could be improved slightly for better readability. 

Visually grouping the scales (perhaps with subheadings or shaded rows) would enhance clarity for quick reading. But it is a minor suggestion.

Comments on the Quality of English Language

Level of English is appropiate. However, consider to avoid redundancies or repetition.

Author Response

Reviewer: 1

Dear Reviewer,

Thank you very much for the peer review of the manuscript “Engaging leadership reduces quiet quitting and improves work engagement: evidence from nurses in Greece”. Thank you for your comments, which have improved the quality of the manuscript. We have addressed all the comments (highlighted with track changes system) in the revised text. Also, we make changes in the manuscript according to the other Reviewers’ instructions.

Please, find below an item-by-item answer to your comments. Hoping the revised manuscript fulfils the journal’s standards, we thank you for your courtesy.

We are looking forward to your response.

Yours sincerely,

The authors

Comments to the Author

The manuscript addresses a topic of interest to the work environment of nurses. It concludes that engaging leadership reduces quiet quitting and improves work engagement among nurses, which is a significant finding.

While not ideal for generalizability, the approach is according to exploratory or early-stage studies and is appropriate given the study’s novelty. 

Comment

Perhaps table formatting could be improved slightly for better readability. 

Visually grouping the scales (perhaps with subheadings or shaded rows) would enhance clarity for quick reading. But it is a minor suggestion.

Response: Done

Dear Reviewer, we group the scales in Table 1 as follows. Please see the Table 1 in text. We choose this approach due to color limitations of the journal.

Engaging Leadership Scale-12

  - Strengthening

- Connecting

- Empowering

- Inspiring

Reviewer 2 Report

Comments and Suggestions for Authors

Dear Author

Thank you for the opportunity to review this paper.

This paper is an ambitious study that attempts to approach the important issues of "quiet retirement" and "work engagement" in the modern nursing field from a new leadership theory called "engaging leadership". The background, research gap, and purpose of the study are clear, and the paper is logically structured as a whole. The methodology is also generally appropriate, and the study is expected to contribute to nursing management and nursing practice.

This paper is meaningful and logically organized, but there is room for further improvement. I hope that my comments will contribute to improving the manuscript.

Point 1

The introduction logically describes the information necessary for readers. It clearly points out the research gap that "there is little research on the effect of engaging leadership on quiet retirement in nurses," highlighting the originality and necessity of this study. The research purpose is also clearly stated, and it is very well structured as an introduction.

Point 2

  1. Materials and Methods will need some revisions. First, regarding Facebook groups for nurses in 2.1 Design and sampling. How large is this group and what is the purpose of creating it? Basic information should be provided, such as the age group, affiliation, and area distribution of the members.

Point 3

Similarly, inclusion criteria are described in 2.1 Design and sampling, but were there any exclusion criteria? For example, were nurses who are already leaders, such as head nurses and department heads, included in the participants?

Point 4

Furthermore, several criteria are described in the sample size calculation, but is there any basis for determining these criteria?

Point 5

How did you determine the demographic variables such as sex in 2.2 Instruments? This overlaps with the previous point 3, but did you not ask about job titles such as head nurses?

Point 6

Regarding the “Engaging Leadership Scale-12” (ELS-12), is this a scale that measures the engagement leadership that nurses themselves have? Or is it a scale that nurses answer by imagining the leader of their own department? To facilitate readers' understanding, we recommend adding an explanation and listing 2-3 specific items.

Point 7

I think it is appropriate to include 3. Results, 4. Discussion, and limitations.

Author Response

Reviewer: 2

Dear Reviewer,

Thank you very much for the peer review of the manuscript “Engaging leadership reduces quiet quitting and improves work engagement: evidence from nurses in Greece”. Thank you for your comments, which have improved the quality of the manuscript. We have addressed all the comments (highlighted with track changes system) in the revised text. Also, we make changes in the manuscript according to the other Reviewers’ instructions.

Please, find below an item-by-item answer to your comments. Hoping the revised manuscript fulfils the journal’s standards, we thank you for your courtesy.

We are looking forward to your response.

Yours sincerely,

The authors

Dear Author

Thank you for the opportunity to review this paper.

This paper is an ambitious study that attempts to approach the important issues of "quiet retirement" and "work engagement" in the modern nursing field from a new leadership theory called "engaging leadership". The background, research gap, and purpose of the study are clear, and the paper is logically structured as a whole. The methodology is also generally appropriate, and the study is expected to contribute to nursing management and nursing practice.

This paper is meaningful and logically organized, but there is room for further improvement. I hope that my comments will contribute to improving the manuscript.

Comment

Point 1

The introduction logically describes the information necessary for readers. It clearly points out the research gap that "there is little research on the effect of engaging leadership on quiet retirement in nurses," highlighting the originality and necessity of this study. The research purpose is also clearly stated, and it is very well structured as an introduction. 

Response: Done

Dear Reviewer, thank you for your kindness.

Comment

Point 2

  1. Materials and Methods will need some revisions. First, regarding Facebook groups for nurses in 2.1 Design and sampling. How large is this group and what is the purpose of creating it? Basic information should be provided, such as the age group, affiliation, and area distribution of the members.

Response: Done

We add the following text in the section 2.1 Design and sampling.

…These groups include approximately 2000 nurses aged 22 to 70 years old that have been working in clinical settings. The purpose of these groups is to provide information on work-related issues regarding nursing, such as scientific articles, laws in health care services and nursing, and employees issues. Nurses from any place in Greece may apply to be a member of these groups. Administrators of the groups examine the application and allow membership…

Comment

Point 3

Similarly, inclusion criteria are described in 2.1 Design and sampling, but were there any exclusion criteria? For example, were nurses who are already leaders, such as head nurses and department heads, included in the participants?

Response: Done

We add the following text in the section 2.1 Design and sampling.

…Leaders nurses such as head nurses and department head nurses cannot participate in our study since they were in a different work status from nurses employees. Also, nurses that do not work in clinical settings (e.g. school nurses) were excluded from our study…

Comment

Point 4

Furthermore, several criteria are described in the sample size calculation, but is there any basis for determining these criteria?

 Response: Done

Dear Reviewer, we consider these criteria due to the statistical methodology guidelines. Please, let us know whether you want any changes.

Comment

Point 5

How did you determine the demographic variables such as sex in 2.2 Instruments? This overlaps with the previous point 3, but did you not ask about job titles such as head nurses?

 Response: Done

We add the following text in the section 2.2 Instruments.

…We measured demographic data of nurses by asking them several questions before they answered our study scales…

…Also, we asked participants if they work as head nurses (no or yes), and if they work in clinical settings (no or yes). Applying our inclusion and exclusion criteria, head nurses and those who do not work in clinical settings were removed from further statistical analysis…

Comment

Point 6

Regarding the “Engaging Leadership Scale-12” (ELS-12), is this a scale that measures the engagement leadership that nurses themselves have? Or is it a scale that nurses answer by imagining the leader of their own department? To facilitate readers' understanding, we recommend adding an explanation and listing 2-3 specific items.

  Response: Done

We add the following text in the section 2.2 Instruments.

…Nurses answer the ELS-12 by imagining their supervisor of their own department. Example items on the ELS-12 include the following: “My supervisor encourages team members to develop their talents as much as possible”, “My supervisor encourages collaboration among team members”, and “My supervisor gives team members enough freedom and responsibility to complete their tasks”…

Comment

Point 7

I think it is appropriate to include 3. Results, 4. Discussion, and limitations.

Response: Done

We include 3. Resu

Reviewer 3 Report

Comments and Suggestions for Authors

Dear Authors,

Thank you for your submission. I carefully reviewed your manuscript and provided detailed comments and suggestions to help improve the scientific rigor, clarity, and overall structure of the paper.

Kindly pay particular attention to the referencing, methodology clarity, and data presentation standards.

The current similarity index of the manuscript is quite high (69%), which raises concerns about potential issues related to originality and academic integrity.

I strongly recommend that the authors revise the manuscript thoroughly to reduce the similarity rate. This can be achieved by properly paraphrasing the content, avoiding excessive use of direct quotations, and ensuring that all sources are appropriately cited.

Reducing the similarity rate is essential for the manuscript to be considered for further review and potential publication.

I hope my feedback will be helpful as you revise your manuscript.

Best regards,

Author Response

Reviewer: 3

Dear Reviewer,

Thank you very much for the peer review of the manuscript “Engaging leadership reduces quiet quitting and improves work engagement: evidence from nurses in Greece”. Thank you for your comments, which have improved the quality of the manuscript. We have addressed all the comments (highlighted with track changes system) in the revised text. Also, we make changes in the manuscript according to the other Reviewers’ instructions.

Please, find below an item-by-item answer to your comments. Hoping the revised manuscript fulfils the journal’s standards, we thank you for your courtesy.

We are looking forward to your response.

Yours sincerely,

The authors

Dear Authors,

Thank you for your submission. I carefully reviewed your manuscript and provided detailed comments and suggestions to help improve the scientific rigor, clarity, and overall structure of the paper.

Kindly pay particular attention to the referencing, methodology clarity, and data presentation standards.

The current similarity index of the manuscript is quite high (69%), which raises concerns about potential issues related to originality and academic integrity.

I strongly recommend that the authors revise the manuscript thoroughly to reduce the similarity rate. This can be achieved by properly paraphrasing the content, avoiding excessive use of direct quotations, and ensuring that all sources are appropriately cited.

Reducing the similarity rate is essential for the manuscript to be considered for further review and potential publication.

Response:

We would like to inform you that the manuscript has been published as a preprint (posted February 4, 2025) on the Research Square preprint platform, which is why this percentage appears. Check the link below:

https://assets-eu.researchsquare.com/files/rs-5959549/v1_covered_64cc3e3b-6652-41e7-8d48-b2b8778f3962.pdf?c=1738729910

Also, the editorial office of the journal has not made any comments regarding the issue of plagiarism.

Abstract

Comment

The location where the study was conducted should be mentioned in the Methods section.

Response: done

We rewrite the text as follows.

A cross-sectional study was conducted in Greece with a convenience sample.

Comment

How was the online form shared with the participants?

Response: done

We add the following text.

We utilized Google forms to create an online version of the study questionnaire, which was subsequently shared in Facebook groups for nurses.

Comment

Over what duration were the research data collected?

Response: done

We rewrite the text as follows.

Data collection occurred throughout October 2024 via an online survey.

Comment

What p-value was considered statistically significant (e.g., p < 0.05)?

Response: done

We add the following text.

P-values less than 0.05 were considered statistically significant.

Comment

Keywords should be listed in alphabetical order.

Response: done

We rewrite the text as follows.

Keywords: engaging leadership; leadership; nurses; quiet quitting; work; work engagement

Introduction

The following statements require appropriate referencing:

o The nursing profession, ... quality of healthcare services.

o Nurses frequently lack the ... these demands.

o They are fervent about their job, ... from it.

o Nurses' job engagement is ... carry out their duties. 8-10.

o By embracing the transformational leadership ... by their conduct.

o Simultaneously, they foster ... for their growth.

o The term "quiet quitting" gained prominence ... contemporary work culture.

o According to the ... psychologically detached from their job.

Please review the use of references throughout this section.

Response: done

We added appropriate references to all above statements, except the statement “Nurses' job engagement is ... carry out their duties”, to which references 8-10 correspond.

Materials and Methods

Comment

How many months did the data collection process last?

Response: done

We rewrite the text as follows.

Data collection occurred throughout October 2024 via an online survey.

Comment

On which platform was the online survey created?

Response: done

We rewrite the text as follows.

Specifically, we utilized Google forms to create an online version of the study questionnaire, which was subsequently shared in Facebook groups for nurses.

Comment

Was voluntary participation not emphasized in the inclusion criteria?

Response: done

We add the following text.

…and (d) participation was voluntary..

Comment

What was the calculated sample size?

Response: done

We add the following text.

We employed G*Power version 3.1.9.2. for sample size calculation. In particular, we considered (a) a low effect size (f2=0.05) between engaging leadership, quiet quitting and work engagement, (b) the number of independent variables (four predictors and five confounders), (c) a confidence level of 95%, and (d) a margin of error of 1%. After all, our sample size was estimated at 370 nurses.

Comment

Was the sample size sufficient?

Response: done

We add the following text.

We employed G*Power version 3.1.9.2. for sample size calculation. In particular, we considered (a) a low effect size (f2=0.05) between engaging leadership, quiet quitting and work engagement, (b) the number of independent variables (four predictors and five confounders), (c) a confidence level of 95%, and (d) a margin of error of 1%. After all, our sample size was estimated at 370 nurses.

Comment

How many items were included for demographic variables?

Response: done

We add the following text.

Thus, we included seven items for demographic variables.

Comment

Who developed the Engaging Leadership Scale-12 and in what year? Who validated and tested its reliability, and when?

Response: done

We add the following text.

…Schaufeli developed the ELS-12 in 2021….

… Katsiroumpa et al. validated the ELS-12 in Greek and tested its reliability in 2024…

Comment

Who developed the Quiet Quitting Scale and in what year? Who validated and tested its reliability, and when?

Response: done

We add the following text.

…Galanis et al. developed the QQS in 2023…

… Galanis et al. validated the QQS in Greek and tested its reliability in 2024…

Comment

Please explain the abbreviation "IBM SPSS 21.0".

Response: done

We add the following text.

…IBM SPSS 21.0 (International Business Machines Corporation Statistical Package for the Social Sciences)…

Comment

Instead of stating "mean age was 41.1 years (SD; 9.9 years)", consider using the format: "41.1 ± 9.9 (min max: ... ... )".

Response: done

We rewrite the text as follows.

…In our sample, mean age was 41.1 ± 9.9 years (min, max: 23, 61 years)…

… Mean years of clinical experience was 16.7 ± 14.0 years (min, max: 0.5, 40 years)…

… Descriptive statistics for the study scales are shown in Table 1. Regarding engaging leadership, scores on empowering (mean; 3.04 ± 1.01), connecting (mean; 2.96 ± 1.10), and strengthening (mean; 2.90 ± 0.98) were higher than inspiring (mean; 2.50 ± 1.08).

Regarding quiet quitting, mean score on factor “lack of motivation” (mean; 2.94 ± 1.00) was higher than mean score on factors “lack of initiative” (mean; 2.38 ± 0.92), and “detachment” (mean; 2.14 ± 0.78). In our sample, 66.6% (n=269) were considered as quiet quitters, while 33.4% (n=135) were considered as non-quiet quitters.

Mean score on UWES-3 was 3.52 ± 1.60 with a range from 0 to 6, indicating a moderate level of work engagement….

Comment

Particularly for scale scores, please express them as Mean ± Standard Deviation, and add minimum and maximum values alongside these figures.

Response: done

We rewrite the text as follows.

Descriptive statistics for the study scales are shown in Table 1. Regarding engaging leadership, scores on empowering (mean; 3.04 ± 1.01, min-max; 1-5), connecting (mean; 2.96 ± 1.10, min-max; 1-5), and strengthening (mean; 2.90 ± 0.98, min-max; 1-5) were higher than inspiring (mean; 2.50 ± 1.08, min-max; 1-5).

Regarding quiet quitting, mean score on factor “lack of motivation” (mean; 2.94 ± 1.00, min-max; 1-5) was higher than mean score on factors “lack of initiative” (mean; 2.38 ± 0.92, min-max; 1-5), and “detachment” (mean; 2.14 ± 0.78, min-max; 1-4.75). In our sample, 66.6% (n=269) were considered as quiet quitters, while 33.4% (n=135) were considered as non-quiet quitters.

Mean score on UWES-3 was 3.52 ± 1.60 (min-max; 0-6), indicating a moderate level of work engagement.

Discussion

The following statements should be supported with references:

o These actions encompass the establishment ... development.

o Another suggested action is ... these supportive work conditions.

o The aforementioned recommended supportive actions ... particularly in

nursing administration.

o The management of health organizations ... including nurses.

o Administrations ... the well-being of human resources.

Response: done

We added appropriate references to all statements.

References

For citations such as Gün, Í, please note:

o Since the manuscript is in English, attention should be paid to the usage of…………

Response: done

We corrected the reference in accordance with the English language.

Other Suggestions

Comment

A separate Limitations section should be included in the manuscript.

Response: done

Please, see the text.
